# A Qualitative Study of Rural and Remote Australian General Practitioners’ Involvement in High-Acuity Patients

**DOI:** 10.3390/ijerph20054548

**Published:** 2023-03-03

**Authors:** Sinead Turner, Vivian Isaac, David Lim

**Affiliations:** 1Riverland Mallee Coorong Local Health Network, Berri, SA 5343, Australia; 2School of Allied Health, Exercise and Sports Sciences, Faculty of Science and Health, Charles Sturt University, Albury, NSW 2640, Australia; 3Translational Health Research Institute, School of Health Sciences, Western Sydney University, Campbelltown, NSW 2560, Australia

**Keywords:** rural health services, family physicians, primary healthcare, patient acuity, emergency medical services

## Abstract

This study aimed to understand the experiences, barriers, and facilitators of rural general practitioners’ involvement with high-acuity patients. Semi-structured interviews with rural general practitioners in South Australia who had experience delivering high-acuity care were audio-recorded, transcribed verbatim, and analyzed through content and thematic approaches incorporating Potter and Brough’s capacity-building framework. Eighteen interviews were conducted. Barriers identified include the inability to avoid high-acuity work in rural and remote areas, pressure to handle complex presentations, lack of appropriate resources, lack of mental health support for clinicians, and impacts on social life. Enablers included a commitment to community, comradery in rural medicine, training, and experience. We concluded that general practitioners are a vital pillar of rural health service delivery and are inevitably involved in disaster and emergency response. While the involvement of rural general practitioners with high-acuity patients is complex, this study suggested that with the appropriate system, structure and role supports, rural general practitioners could be better empowered to manage high-acuity caseloads locally.

## 1. Introduction

The Hart’s inverse care law [1] remains true in contemporaneous rural and remote areas [2]. For instance, the total disease burden rates in remote and very remote areas of Australia were 1.4 times higher than in major cities, with very remote areas reporting 2.5 times higher rates of potentially preventable hospitalizations than urban areas [3]. Causes are complex and nuanced but include lower socioeconomic status, increased frequency of trauma, and poorer access to healthcare services [4,5,6,7,8,9]. It is estimated that 7–40% of emergency department presentations are potentially avoidable if patients have enhanced access to appropriate primary healthcare services [10,11,12,13,14,15,16]. Furthermore, previous research has identified the surge capacity of primary healthcare in attenuating the demands on emergency medical services during disasters by managing higher acuity caseloads [17,18]; this is seen in some countries during the response phase to the COVID-19 pandemic where primary healthcare practitioners were utilized to attenuate the demands on acute care services [13]. While there are several definitions of high-acuity care, the consensus is that high-acuity patients require greater monitoring, resource allocation, and time-dependent medical intervention [19,20,21]. In urban and suburban areas, high-acuity patients are often treated in tertiary hospitals by multidisciplinary teams of specialists and clinicians; however, in rural and remote towns in some countries such as Australia and Canada, emergency departments are often staffed by rural general practitioners [4,6,22]. Furthermore, as the perceived leader in the community where rural general practitioners serve [18], they are often called upon to respond when disasters confront their communities [17]. Thus, this study aims to understand the lived experiences of rural general practitioners’ involvement with high-acuity patients and identify barriers and enablers to inform future rural medical workforce recruitment and retention, and health service planning.

## 2. Methods

This qualitative descriptive study [23] used semi-structured interviews with a purposive sample of vocationally registered general practitioners in rural and remote South Australia who had first-hand experience delivering high-acuity care in routine practice, such as being credentialed to staff rural emergency departments and as part of disaster (including natural, manufactured, pandemic) or other incidents of mass trauma response. Participants were recruited using open-source advertisement and promotion by professional peer bodies focused on rural and remote health in South Australia from November 2019 to March 2020. Secondary to this, purposeful snowballing methods were used following initial interviews to recruit participants from underrepresented demographic groups, such as general practitioners working in solo practices and more remote areas. Based on previous studies [5,13,17,18], we concluded that the sample size of 18 was sufficient based on dense sample specificity and the quality of dialogue for this pilot work [24].

Interviews were undertaken in person at the participant’s residence, place of work, or over the phone (during the acute phases of the COVID-19 pandemic). Each interview averaged 40 min in length. Participants were asked questions decided a priori, and face validity of the interview guide was established with two rural general practitioners not involved in the interviews. Questions included their experience working in rural or remote South Australia, motivations, and perceived barriers and facilitators to engaging with high-acuity patients in this rural and remote setting, and types of high-acuity caseloads encountered. The interviews were audio recorded and transcribed verbatim for analysis. Within a month following the individual interview, the respective transcription was returned to the participant for member checking to ensure the accuracy and truthfulness of the data. Four participants provided further written responses which were incorporated into the data analysis.

Data collection and analysis were undertaken concurrently. Following each interview, thematic analysis with inductive data coding was undertaken [25]. The conventional content analysis whereby the coding categories are derived directly from the text data was employed for this study, thus allowing the narrative to emerge from the raw data. Each interview transcript was read several times by the primary researcher (ST) with input from supervisors (DL and VI). Once read in its entirety at least three times, codes from each interview were entered into an Excel spreadsheet. The codes were then compared between interviews to identify themes. These themes, including deidentified data, were then presented and discussed with an informed insider. Potter and Brough’s capacity-building framework [26], which describes nine interdependent components comprising four hierarchies to support systemic capacity building, was applied for theoretical triangulation.

To ensure reflexive practice, the primary researcher (ST) recorded introspective notes throughout the data collection and analysis stages, with specific reflection immediately following each interview, after initial transcript analysis, and before transferring codes to Excel. These notes were discussed periodically with the project supervisor (DL) to explore the researcher’s assumptions and perspectives.

## 3. Results

A total of 18 participants (P01-P18) were recruited for the study; their demographics can be found below (Table 1). Participant demographics were comparable to Australian rural and remote general practitioners based on gender and rurality [27]. Six themes were identified.

### 3.1. Commitment to Community

Commitment to the community where the participants worked and often lived was the primary driving force behind rural general practitioners’ involvement with high-acuity patients.


*It’s almost like becoming a monk, I made a commitment to this community*
(P03).


*Doing afterhours high-acuity emergency stuff is simply not at all financially rewarded. The only reason I think people still do it is out of a sense of ethical obligation or… duty of care to their community*
(P07).

Several factors played into this, including reference to rural residents deserving the same care availed to their urban counterparts. The benefit of rural general practitioners’ involvement in high-acuity cases was also highlighted by participants in that it offered a unique opportunity for comprehensive care in their community, thereby fulfilling a belief that caring for the community creates a more robust municipality.


*We are there for the patient longitudinally. We’re there for them, not only for their critical illness, but we’re also the ones looking after them when they get back*
(P09).

### 3.2. Inability to Avoid High-Acuity Work

One of the barriers associated with committing to servicing their community was reports of participants feeling unable to avoid providing higher acuity care. This pressure primarily came from the community. A staggering number of participants reported being called to emergencies when not on-call or being “thrust into the doctor role afterhours” while in the community.


*We live in our communities so we’re pretty much 24/7 on call anyway*
(P09).


*When you’re living in a rural area, you’re the one that gets called in*
(P10).


*My daughter plays at their netball club… and if someone gets knocked unconscious or has a neck injury then I’m the person that people look towards, and so I have to be prepared to step up to that*
(P16).


*I’m the only doctor around the place, I haven’t got too much choice. Anything that comes in I’ve got to be able to handle*
(P18).

This inability to avoid high-acuity work was also discussed in the context of perceived pressure from the community, peers, and employers to be involved with high-acuity patients. Some participants reported pressure to handle presentations they deemed too complex for their current experience or seniority. The discrepancy between what participants felt they could handle and the reality of what they were dealing with was also a key issue focused on, primarily by less experienced and predominantly female participants.


*There is a real sort of expectation in lots of directions that you can manage these cases that you really don’t feel that confident in managing … There’s a lot of expectation to be able to manage quite complex presentations*
(P15).

Another effect of rural general practitioners staffing the local emergency department, referred to as being “on-call”, is that many participants deemed the community expectation and pressure to be excessive. This was linked to increased burnout rates due to irregular sleep patterns and financial burdens, as many found it impossible to work their full hours following an overnight on-call shift.


*I do sometimes find it overwhelming, particularly if I haven’t had the chance to eat or go to the toilet or to sleep for that matter. Which seems to be the case every on-call*
(P08).


*The long hours of emergency and the number of presentations that we have… I did struggle with that quite a bit*
(P13).

### 3.3. Negative Impact on Rural General Practitioners

The anxiety, overwhelm, and guilt coupled with the above issues regarding the inability to avoid high-acuity work, dealing with presentations perceived to be too complex, and burnout were confounded by concerns regarding available supports focusing on rural general practitioners’ mental health. Factors such as previous negative encounters including patient deaths or medicolegal cases, lack of security presence, particularly when handling acutely mentally unwell patients, and the occasional nature of high acuity in rural settings being generally stressful were identified as contributors to poor mental health by the participants.


*I wouldn’t be surprised if a lot (of) people have stopped doing this job from PTSD [posttraumatic stress disorder] or from severe anxiety. That’s something that perhaps is a little unspoken… assessing high-acuity cases when you don’t work in the area as your main job, so you’re not working in emergency full time, is incredibly stressful and takes a huge emotional toll*
(P02).

These issues were compounded by the perceived lack of mental health support offered to rural general practitioners in contrast to other health professionals working with high-acuity patients, such as paramedics. Some participants identified coping strategies, including detachment, debriefing with colleagues, and accessing mental health support services, to be beneficial.


*If you were really mentally struggling there is a group in South Australia called Doctors for Doctors, but again you have to know about it, you have to be willing to access it*
(P02).


*When I’m working, I switch off… I can detach myself*
(P03).

Participants also discussed the negative impact on-call rosters have on their social lives, with particular emphasis on reduced availability to spend time with family.


*It does affect your social life, and your family life, and your everything else when you’ve gotta (go), ‘oh no, I can’t do that cause I’m on call’*
(P04).


*If [doctors] are thinking about sort of work life balance issues, they’re thinking about having a family, they’re thinking about having hobbies, they’re thinking about socializing with their friends who have remained in the in the big city, people aren’t wanting to find themselves on-call all the time for just everything*
(P14).

This impact was also discussed in terms of restricted social interactions while on-call, namely the need to abstain from alcohol, to be close to the hospital, and to ensure opportunities to sleep were availed.


*That’s also why I don’t like emergency, like on-call at the hospital, because it intrudes into my personal life*
(P15).

### 3.4. Comradery in Rural Medicine

A supportive factor spawned from the reportedly stressful nature of high-acuity work in rural health settings was the strong relationships with colleagues within the participant’s town and between rural intrastate towns. This comradery manifested in the pooling of physical resources between local towns, but more broadly with sharing of time and advice over the phone.


*That’s something that is just part of that personality and I think why rural doctors are really awesome people, because they are… willing to share their experience and time, and help you out*
(P05).

This sense of support, particularly from more experienced local rural general practitioners, was an enabler for ongoing involvement with high-acuity cases and a strong motive for persisting despite the above-outlined concerns.


*I’m part of a team, and… I have to pull my weight. So that would be my reason for doing it*
(P10).

### 3.5. Lack of Support

Lack of support was the most common recurrent theme amongst the participants, with a particular focus surrounding the fear of being the only doctor present during an emergency, which participants noted could be highly stressful.


*It can be extremely stressful being the only doctor in a country hospital where there’s 1 RN (registered nurse) and 1 EN (enrolled nurse) if an emergency comes in*
(P16).

This idea was echoed in more general high-acuity settings with participants noting they preferred teamwork, which at times led to reports of participants preferencing larger practices where staff were more abundant.


*I would feel a whole lot more comfortable if there was somebody else there to bounce ideas off of*
(P13).


*An awful lot of young doctors didn’t want to come out and do this sort of thing and just wanted to go to large practices where… (they) have a hospital right there and lots of staff*
(P18).

Mitigation of this need for a larger local team came in the form of strong external support such as retrieval services like MedSTAR and the Royal Flying Doctors Service in rural Australia, as well as from tertiary hospitals in metropolitan areas such as the South Australian Virtual Emergency Service, a telemedicine option for lower triaged patients to be seen by an external medical practitioner.


*Knowing that there is support at the end of a phone, either within your town or in Adelaide… or wherever you can ring I think is really important*
(P06).

Well-trained, familiar triaging nurses were identified as a strong protective factor. In particular, many participants reported personal knowledge of the nurse providing handover or triage as a pivotal consideration regarding the clinician’s confidence in the management of higher acuity caseloads.


*Knowing there’s a good nursing team, so if the nurse’s called me up… knowing who’s telling me the story makes me think okay I’m more comfortable with that because I know who they are and what their skills are*
(P05).

### 3.6. Skills and Training

The frequency of skill use and self-perceived competence was a contentious issue among participants. A larger proportion of predominantly lesser experienced participants reported the lower volume of high-acuity presentations in rural settings as a barrier to maintaining skills and confidence.


*I’ve done all the basics at the courses and stuff, but I don’t use it enough… doing it once a year I don’t think would make me competent to do it*
(P12).


*For people to become comfortable with high-acuity cases then it really is about familiarity and training. There really is no substitute for that*
(P07).

This was raised as an area of concern, with some participants reporting fear of causing more harm by implementing skills only previously practiced in simulation settings, or most recently performed months to years prior.


*Am I going to stuff that up because I haven’t intubated anybody since medical school… if I decided to intubate somebody then I run the risk of causing more harm than if I just left them alone*
(P15).

This idea was countered by more experienced doctors who felt that ‘occasionalism’ was a defining skill of rural generalists, and that mastery of a skill early in training resulted in reduced need for high volumes of repetition of said skill to maintain competence.


*I think the infrequency with which you deal with it is not a big deal, and I think if you dig away a little bit deeper you will see that GPs (general practitioners) actually are used to make hard decisions and dealing with high stress and high stakes situations … I might only put in two or three chest drains a year, but I’m actually confident in being able to do this only two or three times a year, because we’re actually quite used to infrequency and uncertainty, that’s probably our defining skill in rural generalism*
(P09).

In addition, more experienced participants commented that rural high-acuity work is easier currently than it was before, with one proposed modality for this disparity being less exposure to rural practice during medical training and an overreliance on specialists.


*Student’s first exposure to medicine is of course in tertiary hospitals where there are so many different specialties… and so everything is differentiated into different parts and the students don’t get used to the idea that you just do it all to a certain level that you train to*
(P14).


*I believe our education and training system from university through to residency, internship, seems to be designed to make people systematically feel underconfident. They’re all taught to feel they couldn’t possibly do that unless they had all this extra experience, all this extra training, lots of specialists on their beck and call immediately and so on*
(P18).


*When we were doing our junior doctor positions, we were doing 60, 70, sometimes 100 h a week work, so that we were getting exposed to a lot more clinical scenarios and clinical cases… I think it is a little bit harder for the doctors who are coming out of the system at the moment, because they may not have seen the same clinical stuff that we did when we did our training*
(P17).

This difference was also felt by younger participants, with some reporting concerns regarding the loss of older, more practiced clinicians and the effect this loss of experience has on remaining practitioners.


*Several rural areas miss out on the experience that the older doctors are able to give*
(P13).

Identified barriers to engaging in meaningful training included the distant location of training, inability to take leave from work, and costs associated with attending training sessions, as well as time lost while training. Adequate remuneration, locum support, and place-based training were identified as enablers in undertaking training.


*I find the ones that are based here as more useful than the ones that I have to travel to, mainly just because when it’s based here I’m also getting those skills about what equipment do we have here, where is all of that*
(P13).

## 4. Discussion

In many countries such as Australia and Canada, where rural general practitioners are often called on to respond to high-acuity caseloads either in a routine capacity, such as credentialed for the local emergency medical service [6], or ad hoc, such as being a clinician in the local rural setting, rural general practitioners have been reported to play a pivotal role in all stages of rural emergency and disaster prevention, preparedness, response, and recovery [7,13,14,17,18,29]. The capacity for rural general practitioners to be involved in high-acuity caseloads is complex, especially as there is currently a limited understanding of the scope and involvement of general practitioners in emergency and disaster management [17], and general practitioners may be overlooked, underutilized, or poorly utilized in crisis and disaster management. A survey conducted during the first wave of the COVID-19 pandemic (April–May 2020) of 1035 respondents from 111 countries found that primary healthcare was not effectively incorporated nor integrated with the public health response to the pandemic [30]. Yet, the successful integration of primary healthcare clinicians during a pandemic and other disasters has the potential to attenuate the demands on acute care services [7,13,18].

This research started prior to the formal declaration of COVID-19 in March 2020, to understand the experiences of rural general practitioners in managing high-acuity caseloads either in their routine capacity or ad hoc. The context for this research was the implementation of the Australian Government’s Stronger Rural Health Strategy [31] and the Australian General Practice Training Program Rural Generalist Policy 2020 to build greater capacity in rural health across Australia. Medical practitioners who are trained as rural generalists are expected to provide expert medical care in all rural contexts, including the provision of high-acuity emergency care, and be involved in disaster planning [32]. Thus, a planned outcome of this research is to inform the policy discourse, medical workforce, and health service planning on how to build systemic capacity to integrate rural generalists in emergency and disaster management. This research was able to compare the participants’ response prior to COVID-19 and in the immediate response phase to the pandemic. Data were integrated to synthesize the unique narrative of the frontline rural and remote general practitioners’ experience so as to understand their scope and involvement during a crisis. To maintain research rigor, Potter and Brough’s capacity-building framework [26] was employed for theoretical triangulation.

This research highlighted the strong sense of moral duty rural general practitioners felt they owed to the people and community they cared for, and their heightened concerns surrounding anything which diminished their ability to practice safely. Less experienced participants appeared to embody these concerns the most and voiced feelings of being overwhelmed and isolated, particularly during high-pressure scenarios such as during emergency response work. Conversely, more experienced participants found current scenarios a relief compared to their early training years with telemedicine and external supports more readily available, as well as finding security in their experience and appreciation of their own and system limitations.

Whilst there is a strong role capacity for rural general practitioners to be involved in emergency and disaster management, other domains of systemic capacity would need to be considered to facilitate greater utilization of current and emerging rural generalists in the management of high-acuity caseloads. For instance, meaningful staffing coupled with a tailored individualized approach to local facility capacity based on local needs [6], integration of enabling technology with individual personal and performance capacity, and reasonable access to regional or national supervisory support could improve technical efficiency [4].

COVID-19 has seen an unprecedented increase in the use of telemedicine in healthcare delivery, which appeared to be embraced by rural general practitioners. System enablers from the Australian Government for the technology uptake included the introduction of 56 Medicare Benefits Schedule telemedicine item numbers for general practitioners. Benefits of telemedicine include increased access to healthcare, reduced travel costs and employment disruption, reduced patient non-attendance rates, improved process efficiency in general practice (for example, repeat prescription or reporting of pathology results), and improved practitioner confidence due to access to specialist consultation. In Australia, the Royal Australian College of General Practitioners and the Australian College of Rural and Remote Medicine both provide guidelines and resources on conducting tele-consult and training in telemedicine systems. With telemedicine now a permanent part of the Australian general practice toolkit, rural general practitioners are another step closer to better supporting their rural communities in times of crisis by being able to take on higher acuity caseloads when required.

Limitations of this study include the restriction of the location to rural and remote South Australia and the sample size of 18 general practitioners. Factors that influenced this decision include the declaration of the COVID-19 pandemic in February/March 2020 which impacted recruitment. Furthermore, this study focused on acuity levels rather than specific types or sources of high acuity (e.g., road traffic accidents); whilst this is congruent with Holzemer’s Outcomes Model for Health Care Research [33] and other research on high acuity, combining different types or sources of high-acuity caseloads may reduce the sensitivity of the synthesized narrative to a particular high-acuity caseload.

## 5. Conclusions

While the involvement of rural general practitioners with high-acuity patients is complex, this study suggested that rural general practitioners could be better empowered to manage high-acuity caseloads locally. The findings highlighted contributing factors such as feeling unsupported in the workforce, self-perceived issues around competence and confidence, and on-call roster demands as barriers to involvement, while a strong sense of community was an important facilitator for engagement. These findings should be considered when planning for the greater integration of rural generalists in the provision of high-acuity care.

## Figures and Tables

**Table 1 ijerph-20-04548-t001:** Characteristics of study participants (*n* = 18).

Gender	*n* (%)	Reference [27]
Male	10 (56%)	5330 (58%)
Female	8 (44%)	3828 (42%)
Rurality (ASGC-RA) ^1^		
Inner regional	10 (56%)	1150 (53%)
Outer regional	6 (33%)	711 (33%)
Remote	2 (11%)	204 (9%)
Very remote	0 (0%)	102 (5%)
Experience in rural (years)		
1–10 years	6 (33%)	6442 (77%)
10–20 years	5 (28%)	1037 (13%)
20 + years	7 (39%)	850 (10%)
Practice style		
Group	14 (77%)	7761 (85%)
Solo	3 (17%)	591 (7%)
Locum	1 (6%)	210 (2%)

^1^ ASGC-RA = Australian Statistical Geographical Classification-Remoteness Area [28].

## Data Availability

The data presented in this study are available on request from the corresponding author. The data are not publicly available due to the conditions of institutional ethics approval.

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
