# Peer review of "A Qualitative Study of Rural and Remote Australian General Practitioners’ Involvement in High-Acuity Patients"

_ijerph, 2023, doi:10.3390/ijerph20054548_

Round 1
Reviewer 1 Report
The authors are to be commended for a well-documented, carefully analyzed qualitative study of issues faced by rural General Practitioners.
This reviewer's only hesitation comes in the Discussion section. Issue: techniques for supporting rural practitioners. Full disclosure: writing from the United States. In the US, there are multiple programs, notable Project ECHO, that attempt to use tele-medicine techniques to link rural physicians to urban centers for grand rounds, clinical presentation of complicated cases being managed in the rural setting, and so on. Not being familiar with the broadband situation in Australia I cannot comment on the suitability of such programs for that setting, but it seems like something the authors could pursue that might influence their recommendations.
Author Response
Dear Reviewer, thank you for the encouragement and the mention of the ECHO program. In our previous work on hepatitis C, we have indeed modelled aspects of ECHO in the program delivery, thank you for helping us make the connection. We have now made reference to tele-medicine in our Discussion section as recommended.
Reviewer 2 Report
The paper at hand is about a very timely topic. From the scientific point article is rather weak so it could be publish as an professional article. The aim of the research is not clearly explained.
Rows 62-62 : (Methods) - In my opinion the sample size of 18 is too small. My question: is 18 the minimum sample size? Can I take larger sample?
Rows 75-76: "Data collection and analysis were undertaken concurrently. Following each interview, thematic analysis with inductive data coding was undertaken". Could you be more specific?
Author Response
Dear Reviewer, thank you for the constructive comments. We have now provided further justification and rationale for undertaking this research, we have also addressed the matter with regard to the sample size for qualitative research and provided more information about the data analysis process.
Reviewer 3 Report
The manuscript by Turner et al., investigated the experiences, barriers, and facilitators of rural GPs involved with high-acuity patients. They implemented a survey-based questionnaire to identify barriers that rural GPs face and their perspectives on how their role supports the community, and what is needed to better empower them to manage high-acuity caseloads. The introduction is well-structured with background information and points out the gap in the literature. The methods were sound and appropriate. One point to note is that authors can simplify how they came to the sample size number of 18 without just providing the reference (#24). The findings are considered interesting. The methods include all necessary elements and are very well described. The discussion mentions the novelty of the study and how this study could potentially advance the field. The authors have also suggested multiple potential improvement strategies for the betterment of the GPs utilization in rural areas. The paper is overall well-written and easy to follow.
Author Response
Dear Reviewer, thanking you sincerely for the encouragement. This is a coursework student project and your positive comments have helped affirm the student's commitment to further research training and embedding this in her clinical practice. We have addressed the constructive feedback for improvement, especially with regard to the sample size in the revised manuscript.
Round 2
Reviewer 2 Report
Corrected article. No comments.